# The Change in Whole-Genome Methylation and Transcriptome Profile under Autophagy Defect and Nitrogen Starvation

**DOI:** 10.3390/ijms241814047

**Published:** 2023-09-13

**Authors:** Yunfeng Shi, Baiyang Yu, Shan Cheng, Weiming Hu, Fen Liu

**Affiliations:** 1Lushan Botanical Garden, Jiangxi Province and Chinese Academy of Sciences, Jiujiang 332000, China; shiyf@lsbg.cn (Y.S.); 12216011@zju.edu.cn (B.Y.); chengshan@lsbg.cn (S.C.); 2Department of Agronomy, College of Agriculture and Biotechnology, Zhejiang University, Hangzhou 310058, China

**Keywords:** autophagy, nitrogen starvation, whole-genome bisulfite sequencing (WGBS), RNA-seq, hormone

## Abstract

Through whole-genome bisulfite sequencing and RNA-seq, we determined the potential impact of autophagy in regulating DNA methylation in *Arabidopsis*, providing a solid foundation for further understanding the molecular mechanism of autophagy and how plants cope with nitrogen deficiency. A total of 335 notable differentially expressed genes (DEGs) were discovered in wild-type *Arabidopsis* (Col-0-N) and an autophagic mutant cultivated under nitrogen starvation (*atg5-1*-N). Among these, 142 DEGs were associated with hypomethylated regions (hypo-DMRs) and were upregulated. This suggests a correlation between DNA demethylation and the ability of *Arabidopsis* to cope with nitrogen deficiency. Examination of the hypo-DMR-linked upregulated DEGs indicated that the expression of MYB101, an ABA pathway regulator, may be regulated by DNA demethylation and the recruitment of transcription factors (TFs; ERF57, ERF105, ERF48, and ERF111), which may contribute to the growth arrest induced by abscisic acid (ABA). Additionally, we found that DNA methylation might impact the biosynthesis of salicylic acid (SA). The promoter region of ATGH3.12 (PBS3), a key enzyme in SA synthesis, was hypomethylated, combined with overexpression of PBS3 and its potential TF AT3G46070, suggesting that autophagy defects may lead to SA-activated senescence, depending on DNA demethylation. These findings suggest that DNA hypomethylation may impact the mechanism by which *Arabidopsis* autophagy mutants (*atg5-1*) respond to nitrogen deficiency, specifically in relation to ABA and SA regulation. Our evaluation of hormone levels verified that these two hormones are significantly enriched under nitrogen deficiency in *atg5-1*-N compared to Col-0-N.

## 1. Introduction

As a universal mechanism in eukaryotes that promotes cell longevity and nutrient recycling, autophagy helps plants cope with various biotic/abiotic stresses [1]. Autophagy in plants primarily encompasses three types: microautophagy, macroautophagy, and mega-autophagy. In the process of microautophagy, the separation and extraction of components in vacuoles are realized by direct wrapping within vacuole/lysosome membranes. This is related to the accumulation of anthocyanidin in vacuoles and the elimination of damaged chloroplasts in plants [2]. Mega-autophagy is implicated in both programmed cell death (PCD) that occurs during developmental processes and pathogenic invasion [3]. During macroautophagy (hereafter referred to as autophagy), cargos are trapped in newly formed cytoplasmic vesicles, which are created when a cup-shaped phagophore (also known as an isolation membrane) expands and envelops the cytoplasm, eventually sealing off to form the autophagosome, a double-membrane-bound structure. Subsequently, the outer membrane of the autophagosome fuses with the tonoplast, leading to the release of the internal vesicle as an autophagic body [3]. The autophagy pathway is regulated by a sophisticated mechanism for which over 40 ATG genes have been identified in plants [4,5,6,7,8]. In recent years, various studies have indicated that autophagy plays a crucial role in diverse biological processes in plants, including growth, development, degradation of starch, senescence, and control of lipid metabolism [9]. In *Arabidopsis*, autophagy also affects the growth of pollen tubes. Through the specific mediation of ATG8-family interacting motif (AIM) recognition sites, ATG8e and mitochondria can bind and interact spatiotemporally, thereby inducing mitochondrial autophagy [10]. In addition, the relationship between autophagy and epigenetics in animals has also been validated. Under nutritional deficiency, fibroblast growth factor-21 (FGF21) signaling activates the global expression of autophagy-related genes through the histone H3K27-ME3 demethylase Jumonji-D3 (JMJD3/KDM6B), thus mediating lipid degradation [11]. The autophagy regulatory network may also be regulated by DNA methylation. It has been reported that MAP1LC3 is a key component of the core mechanism of autophagy. DNA methylation mediated by the de novo DNA methyltransferase DNMT3A at the MAP1LC3 locus leads to continuous downregulation of the transcription level of MAP1LC [12]. The relationship between autophagic function and epigenetic regulation has been explored in animals. However, it remains unknown whether plant autophagy and/or nitrogen starvation affect whole-genome DNA methylation.

Numerous studies have examined the function of DNA methylation in the abiotic stress response. DNA methylation resulting from abiotic stress is not limited to specific gene regions; it can occur throughout the genome. Moreover, the genes that are susceptible to modification in response to stress are ostensibly linked to methylation regulation [13]. In research conducted by Jiang, C. et al. on the methylation profile of *Arabidopsis* cultivated in soil with high salt concentration, in comparison to the control group, the stressed lineages amassed approximately 45% more differentially methylated cytosine positions (DMPs) at CG sites (CG DMPs), and the majority of these DMPs were observed to be inherited [14]. In *Arabidopsis*, exposure to low levels of Pi results in a significant alteration in the methylation level of the entire genome, which in turn is linked with the regulation of gene expression for responding to Pi starvation [15]. In a study of DNA methylation of the VRN-A1 gene of winter wheat, Abdul Rehman Khan et al. reported methylation at both CG and non-CG sites. Furthermore, they observed that site-specific hypermethylation induced by cold was transmitted through mitosis at non-CG sites [16]. To date, comprehensive research has been conducted on the response of various plants and crops to abiotic stress. This includes studies on the correlation between autophagy and abiotic stress and the regulatory mechanism of DNA methylation in plants or crops for resistance to abiotic stress [17,18,19,20,21].

Based on the above research background, the response of plants to abiotic stress seems to have a profound relationship with DNA methylation, so we hypothesized that DNA methylation may be involved in the response of autophagy-deficient mutants to nitrogen deficiency. However, up to now, no relevant research has been conducted on the relationship between autophagy and DNA methylation in plants and how to regulate gene expression by affecting the DNA methylation level. Thus, in this study, whole-genome bisulfite sequencing (WGBS) was performed on *Arabidopsis* autophagic mutants (*atg5-1*) developing under nitrogen starvation to address relevant concerns, in conjunction with transcriptome sequencing (RNA-seq).

## 2. Results

### 2.1. The Whole-Genome Methylation Level of Autophagic Mutants Was Strongly Affected under Nitrogen Deficiency Conditions

To investigate whether autophagy influences plant growth under nitrogen starvation stress through DNA methylation/demethylation, we performed WGBS and RNA-seq for *Arabidopsis* wild-type (Col-0) and autophagic mutant (*atg5-1*) lines under MS growth conditions (MS liquid culture medium) or nitrogen starvation conditions (MS-N liquid culture medium) (Figure 1A). Over 4G clean reads (paired-end reads) were produced for each sample in the WGBS experiment.

The average mapping ratios for alignment with the reference genome for the different treatment groups were 94.06% (Col-0-MS), 92.50% (Col-0-N), 87.64% (*atg5-1*-MS), and 85.06% (*atg5-1*-N), and the conversion ratios were >99.5% (Appendix A). We then separated DNA methylation into three context types (CG, CHG, and CHH) and detected methylation sites based on the above context types [22,23]. According to the methylation level of common sites in all samples, a principal component analysis (PCA) was performed and showed obvious differentiation between wild-type *Arabidopsis* and the autophagic mutant (Figure 1B).

To characterize the methylation changes under nitrogen starvation conditions and in the autophagy mutant, differentially methylated cytosines (DMCs) and differentially methylated regions (DMRs) were identified in different comparison groups (Col-0-MS vs. Col-0-N, *atg5-1*-MS vs. *atg5-1*-N, Col-0-MS vs. *atg5-1*-MS, and Col-0-N vs. *atg5-1*-N).

In Col-0-MS vs. Col-0-N, there were 351, 252, and 31 upregulated DMRs and 565, 375, and 24 downregulated DMRs in the CG, CHG, and CHH contexts, respectively, and among the comparison groups, the fewest DMRs were identified in Col-0-MS vs. Col-0-N. Apparently, wild-type *Arabidopsis* was not significantly impacted by nitrogen starvation, and relatively obvious differences were observed in the comparison between the wild-type (Col-0-MS) and autophagic mutant (*atg5-1*-MS) under normal conditions (1088, 874, and 602 upregulated DMRs and 6690, 6483, and 3149 downregulated DMRs). Autophagy seems to have a greater influence on the whole-genome methylation level than nitrogen starvation, although autophagy is one of the most important pathways for responding to N stress in plants.

By comparing the DMR numbers of *atg5-1* samples in MS and nitrogen starvation conditions (*atg5-1*-MS vs. *atg5-1*-N), 2868, 3265, and 4213 hypermethylated DMRs and 6052, 7070, and 7603 hypomethylated DMRs were identified in the CG, CHG, and CHH contexts, respectively, showing a dramatic difference in Col-0-MS vs. Col-0-N. Significant differences in DMR number were observed in *atg5-1* genotype samples and wild-type samples growing in MS and nitrogen starvation conditions, which seemingly indicated that in wild-type samples, some compensation effects probably resisted the influence of nitrogen starvation on methylation levels; however, this kind of compensation effect might be inhibited due to autophagy defects. Under the same nitrogen starvation conditions, 863, 698, and 659 hypermethylated DMRs and 10691, 11110, and 6913 hypomethylated DMRs were identified in Col-0-N vs. *atg5-1*-N in the CG, CHG, and CHH contexts, respectively (Appendix A) (Figure 1C). We examined the common methylation site for each treatment group, and the results showed that the overall methylation level of the *atg5-1* sample was significantly lower than that of the wild-type sample, which further suggested that the decrease in the DNA methylation level could be largely attributed to the autophagy defect (Figure 1D).

In summary, nitrogen starvation may not have a significant impact on *Arabidopsis* in terms of methylation. However, when autophagy in *Arabidopsis* produces defects, the impact of nitrogen starvation may be amplified.

### 2.2. Autophagy Stimulates Global Expression of the Arabidopsis Genome under Nitrogen Starvation Conditions

To examine the gene expression changes involved in resistance to autophagy and nitrogen starvation in *Arabidopsis*, transcriptome profiles under different conditions were generated by RNA-seq (Appendix A), and more than 6G clean reads (paired-end reads) were obtained for each RNA-seq sample.

The number of upregulated genes was obviously greater than the number of downregulated genes in the Col-0-MS vs. Col-0-N comparison group, suggesting that global gene expression was activated by nitrogen starvation to maintain growth. In *atg5-1*-MS vs. *atg5-1*-N, there were similar changes in differential gene expression, indicating that in the case of autophagy-related functional defects, nitrogen starvation may still activate the expression of many genes in response. However, a minimum number of DEGs appeared in Col-0-MS vs. *atg5-1*-MS, indicating that autophagy-related functional defects probably mildly impact plant growth under normal conditions (Figure 2A,B).

In Col-0-MS vs. Col-0-N, the genes associated with the integral component of the membrane, the intrinsic component of the membrane, and the defense response were the most significantly enriched among the upregulated DEGs. KEGG analysis showed that genes associated with ABC transporters were significantly enriched in membrane transport-related pathways, suggesting that ABC transporters probably play an important role when *Arabidopsis* suffers nitrogen starvation. The genes associated with the chloroplast thylakoid membrane, the plastid thylakoid membrane, and the photosynthetic membrane were the most significantly enriched among the downregulated DEGs. Interestingly, most of the top 10 GO enrichment annotations of the downregulated DEGs seemed to be related to thylakoids. We performed a KEGG analysis for DEGs annotated in thylakoid-related functional modules. The DEGs were mainly involved in pathways related to photosynthesis, ATPase, and NADH dehydrogenase (Appendix A).

In Col-0-MS vs. *atg5-1*-MS, the upregulated DEGs were extremely enriched in protein self-association, and most of the functional modules in the top 10 enriched GO terms were associated with protein structure, unfolded protein processing, and hydrolase activity. Understandably, in autophagy, degradation, and recycling of unwanted content and misfolded proteins are the main functions (Figure 2C) [24,25]. A KEGG analysis was performed for these DEGs, which showed that the protein processing in the endoplasmic reticulum and tryptophan metabolism pathways were significantly enriched, and the DEGs enriched in these two pathways were associated with heat-shock-protein- and myrosinase-related genes. (Appendix A). GO enrichment analysis was performed for the downregulated DEGs and showed that oxidoreductase activity and the oxidation–reduction process were the most significantly enriched, while other significantly enriched GO terms were mainly related to the binding of substances and NADH dehydrogenase activity (Figure 2D). KEGG results produced based on GO enrichment showed that most DEGs were associated with oxidative reactions and encoded glutathione transferase (Appendix A).

To explore the functions that were influenced differentially between Col-0-N and *atg5-1*-N, we performed GO enrichment analysis of the upregulated DEGs and removed the DEGs that were present in Col-0-MS vs. *atg5-1*-MS. The results clearly showed that most of the top 10 GO terms were related to oxygen levels and the response to hypoxia, suggesting that plants with autophagy defects were more sensitive to oxygen under nitrogen starvation conditions (Figure 2E). Additionally, we annotated the pathways associated with these upregulated genes using KEGG enrichment. The most upregulated DEGs, which were enriched in oxygen-level-related functions, were associated with the amino acid metabolism pathway, while carotenoid biosynthesis, the MAPK signaling pathway, and the biosynthesis of other secondary metabolites were also enriched (Appendix A). Subsequently, exploration of gene function was carried out through GO and KEGG enrichment for the downregulated DEGs, from which, for functional analysis, DEGs that genes were also present in Col-0-MS vs. *atg5-1*-MS were removed. The DEGs associated with the extracellular region, the glucosinolate metabolic process, and the glucosinolate catabolic process were significantly enriched in metabolic pathways and the biosynthesis of secondary metabolites, mainly including genes encoding various enzymes (Figure 2F and Appendix A).

To explore functional changes between *atg5-1*-MS and *atg5-1*-N, the DEGs that were also present in Col-0-MS vs. Col-0-N were removed. The GO results for the upregulated genes were similar to those for Col-0-N vs. *atg5-1*-N. The top 10 GO terms were mainly related to the response to changes in oxygen levels, and the KEGG results for DEGs enriched in the GO terms mentioned above showed that the significantly enriched terms were mainly related to metabolic and biosynthetic pathways of various substances, which mainly involved genes associated with a cytidine/deoxycytidylate deaminase family protein and genes encoding a cytidine deaminase, a gamma-glutamyltransferase and glycosyl hydrolase family protein (Appendix A). For downregulated DEGs, GO enrichment showed that extracellular function and oxygenase activity-related function were significantly enriched. KEGG annotation indicated that the genes with the functions mentioned above were mainly enriched in biosynthesis- and metabolism-related pathways (Appendix A).

### 2.3. Autophagy Deficiency May Induce Downregulation of DNA Methylation Levels, Thus Activating Gene Expression Related to Cell Death and Catabolic Pathways

To investigate the potential influence of DNA hypomethylation induced by autophagy defects, we counted the number of hypo-DMRs associated with upregulated DEGs. The results showed that most of the DEGs were identified in Col-0-MS vs. Col-0-N, but only a few of these DEGs were regulated by DNA methylation, which indicates that DNA methylation is not closely involved in the mechanism underlying the response to nitrogen deficiency in the growth process of wild-type *Arabidopsis* under nitrogen deficiency conditions.

In Col-0-MS vs. *atg5-1*-MS, *atg5-1*-MS vs. *atg5-1*-N, and Col-0-N vs. *atg5-1*-N, there were 38, 73, 142 hypo-DMRs associated with upregulated DEGs, respectively, which was comparable to the number of upregulated DEGs identified in the comparison groups mentioned above. The number of upregulated DEGs regulated by methylation was also considerable (Figure 3A). To explain the relationship between DNA methylation and autophagy and the possible mechanism by which DNA methylation regulates autophagy mutants’ resistance to nitrogen deficiency, we mainly focused on Col-0-MS vs. *atg5-1*-MS and Col-0-N vs. *atg5-1*-N in the analysis process.

GO enrichment analysis of hypo-DMRs associated with the upregulation of DEGs in different comparison groups showed that DNA hypomethylation may have a regulatory effect on nitrogen starvation and autophagy defects. This effect may be controlled by specific genes in different biological pathways. The GO enrichment of hypo-DMRs associated with upregulated DEGs in Col-0-MS vs. *atg5-1*-MS showed that the upregulated DEGs were most significantly enriched in GO terms associated with hydrolase activity and meta-hydroxylase activity, suggesting that DNA hypomethylation significantly regulated hydrolase activity under autophagy defects (Figure 3B). The genes encoding cellulase (glycosyl hydrolase family 5) protein (AT1G13130), glycosyl hydrolase 28 (GH28) family polygalacturonase (PG) protein, cell wall invertase 5, disease resistance protein (TIR–NBS–LRR class) family and tricoumaroylspermidine meta-hydroxylase had the highest fold change. Surprisingly, several hypo-DMR-associated DEGs were enriched in GO terms related to various substance metabolic processes and biosynthetic processes, such as the secondary metabolite biosynthetic process, the lignin biosynthetic process, the lignin metabolic process, and the negative regulation of starch metabolic process, suggesting that metabolic and biosynthetic activity is induced by autophagy defects. Interestingly, the QQS (AT3G30720), ATMYB29 (AT5G07690), XTH6 (AT5G65730), AT4G08160, and BGLU34 (AT1G47600) genes were enriched in multiple metabolism-related GO terms, and in biosynthesis-related GO terms, ATCAD8 (AT4G37990), LAC11 (AT5G03260), and ATMYB29 (AT5G07690) were also enriched. In addition, the DNA methylation processes were probably regulated. For example, DNA methylation on cytosine within a CG sequence, maintenance of DNA methylation, DNA methylation on cytosine, and all GO terms related to methylation were enriched, including only ORTH3 (AT1G57800), which indicates that ORTH3 probably plays an important role in regulating the methylation process induced by autophagy defects (Appendix A).

To explore the functions regulated by the autophagy mutant compared with the wild type under nitrogen starvation, GO enrichment was performed for Col-0-N vs. *atg5-1*-N. The results indicated that hypomethylation greatly affected the response to biotic stimuli (Figure 3B). The DEGs that exhibited the highest fold change were RLP3 (AT1G17250), AT1G32763, AIG1 (AT1G33960), LECRK-III.2 (AT2G29250), AT5G10530 (LECRK-IX.1), AT5G42223, and SAG12 (AT5G45890). In addition, the catabolic processes for many substances were significantly enriched, and five hypo-DMR-related upregulated DEGs, AT4G18350, AT4G32810, AT4G39650, AT5G24540, and AT4G29640, were mainly involved in these processes. AT4G29640 and AT4G39650 exhibited the highest fold change, suggesting that these two genes probably play an important role in resistance when autophagic mutants are subjected to nitrogen starvation stress. This response mechanism may be due to the deficiency of autophagy function, which leads to *Arabidopsis* being unable to maintain the original level of DNA methylation under nitrogen deficiency, thus activating the decomposition of some substances. Meanwhile, the regulatory pathways related to cell death were also affected, and AT2G32460, AT5G10530, AT5G13320, and AT5G45890 were significantly enriched in these GO terms, suggesting that they may be key for establishing the relationship between DNA methylation and premature aging caused by autophagy deficiency. Similar to the catabolic process for many substances, the downregulation of DNA methylation levels may activate gene expression regulating senescence or death in *Arabidopsis*. This regulatory process may be activated by nitrogen deficiency, as there was no enrichment observed for many cell-death-related functions in the Col-0-MS vs. Col-0-N comparison group. Therefore, we speculate that defects in autophagy may affect the level of DNA methylation, and this effect reduces the tolerance of *Arabidopsis* to nitrogen deficiency.

Additionally, compared with the result for Col-0-MS vs. *atg5-1*-MS, the response to organonitrogen compounds was exclusively enriched when *Arabidopsis* was under nitrogen starvation, and it was probably influenced by AT2G14610, AT2G46400, AT3G44350, and AT5G59820, which likely indicates that under nitrogen deficiency, changes in the level of DNA methylation may be involved in mediating the utilization of nitrogen by *Arabidopsis* when the autophagy function has broken down.

We performed Student’s *t*-tests for the TPM (transcripts per million) value of several methyltransferases and demethylases of different comparison groups. A maximum number of methyltransferases and demethylases were differentially expressed in Col-0-N vs. *atg5-1*-N, which indicates that DNA methylation might be regulated by autophagy under nitrogen starvation conditions (Figure 3C).

### 2.4. Nitrogen Deficiency May Induce Transcription Factors of the ERF and C2H2 Type Zinc Finger Families to Regulate ABA-Induced Growth Arrest and SA Biosynthesis in atg5-1 Autophagic Mutants

A Venn diagram was plotted for different comparison groups, and most of the upregulated DEGs were methylated exclusively in Col-0-N vs. *atg5-1*-N (Figure 4A). Hypo-DMR is usually associated with upregulated expression of genes through the recruitment of transcription factors (TFs). It has been reported that low-methylation regions (LMRs) can be used to identify transcriptional enhancers, and TF binding sites can usually be detected in LMRs, which tend to be occupied by cell-type-specific TFs [26,27]. Thus, we predicted TFs that can bind to hypo-DMRs for Col-0-MS vs. *atg5-1*-MS using PlantTFDB, and a total of 53 TFs were identified, which included 17 types of TF families, the most abundant among which were the bZIP, Dof, and GATA families [28,29]. However, the TFs associated with hypo-DMRs did not exhibit significant differences in the expression profiles determined by RNA-seq. In Col-0-N vs. *atg5-1*-N, a total of 26 types of TF families were predicted, the most abundant of which were the ERF, NAC, MYB, C2H2, LBD, GATA, and Trihelix families. By analyzing TF expression, *AT1G02230* (*ANAC004*), *AT2G40340* (*ATERF48*), *AT3G04060* (*ANAC046*), *AT3G12910*, *AT3G46070* (C2H2-type zinc finger family protein), *AT5G18270* (*ANAC087*), *AT5G51190* (*ERF10*5), *AT5G64750* (*ERF111*), and *AT5G65130* (*ERF57*) were shown to be significantly differentially expressed (*p*-value < 0.05; FC ≥ 1.5), and most of them belong to the NAC and ERF families (Appendix A and Figure 4B). Among the predicted genes, TFs showed the possibility of binding to the DMRs in the *AT2G32460* (*MYB101*), *AT2G29250*, *AT1G32763*, *AT5G13320* (*PBS3*) and *AT5G59820* (*RHL41*) promoter regions; GO enrichment analysis showed that these genes are involved in biotic stimulus, cell death and response to organonitrogen compounds, indicating that these functions were probably regulated by hypo-DMR-recruiting TFs.

MYB101 has been confirmed to be associated with ABA-induced plant growth arrest, and transcription factors of the ERF family also play an important role in the mechanism of plant death. Our results predicted that the transcription factors binding to *MYB101* belong to the ERF family, and these transcription factors were also differentially upregulated in the RNA-seq-based differential expression analysis. This seems to indicate that under nitrogen deficiency, the methylation level of the promoter region of *MYB101* is downregulated, and the introduction of ERF family transcription factors binding to this region activates the ABA-induced plant growth arrest process. In addition, the transcription factor prediction results indicated that *AT3G46070* (C2H2 type zinc finger family protein) could bind to *ATGH3.12* (*PBS3*). *PBS3* is the rate-limiting enzyme in the last two steps of SA biosynthesis, which controls plant aging. Therefore, we speculate that autophagic deficiency can cause significant downregulation of the DNA methylation level in the *PBS3* promoter region, thus recruiting *AT3G46070* to combine with this low methylation region to promote SA biosynthesis and control the premature senescence of *Arabidopsis*.

### 2.5. The qRT–PCR Experiments on Key Genes Involved in the SA and ABA Synthesis Pathways Confirmed the WGBS and RNA-seq Analysis Results

Based on our hypothesis, DNA methylation is involved in the process of ABA-induced growth arrest of *Arabidopsis* in Col-0-N vs. *atg5-1*-N. We thus performed qRT–PCR for the DEGs mentioned above (Figure 5A). We speculate that the level of ABA would increase if it was involved in the premature senescence of the autophagy mutant under nitrogen deficiency, and this is also fundamental evidence that ABA-induced senescence is impacted by DNA methylation. Therefore, we also verified the expression levels of key genes related to ABA biosynthesis, and the results also corresponded to the RNA-seq profile and our hypothesis (Figure 5B).

SA biosynthesis was probably also associated with DNA methylation in our WGBS and RNA-seq analysis results, and DNA methylation might affect the expression of ATGH3.12 (PBS3), which participates in the last two steps of SA biosynthesis. We also performed qRT–PCR for the genes involved in the SA biosynthesis pathway, and the results showed that the expression of ATGH3.12 (PBS3) increased dramatically and that of key genes of the ABA biosynthesis pathway, such as ICS1, also changed significantly (Figure 5C).

The qPCR results of key genes suggested that the levels of the hormones SA and ABA were significantly upregulated, consistent with our hypothesis. This confirms the accuracy of the RNA-seq results and indicates that differential DNA methylation is highly likely related to the mechanism by which ABA and SA regulate plant growth arrest or premature senescence (Appendix A).

### 2.6. The Levels of the Hormones SA and ABA in the atg5-1 Mutant Significantly Increased under Nitrogen Deficiency

We hypothesize that the ABA and SA pathways may be regulated by DNA methylation to affect premature plant senescence. Therefore, we measured the levels of SA and ABA. The results of ABA determination showed that the hormone levels were significantly higher in the *atg5-1*-N group than in the Col-0-N group (Figure 6A). The level of SA showed a significant increase in *atg5-1*-N, exhibiting a more than eight-fold change compared with Col0-N (Figure 6B).

Hormonal measurements indicated that deficiencies in autophagy function may reduce the tolerance of *Arabidopsis* to nitrogen starvation, thus directly or indirectly changing the DNA methylation level, thereby triggering an increase in SA and ABA levels and leading to premature aging.

## 3. Discussion

DNA methylation is an epigenetic modification that commonly occurs during plant growth and development and plays an important role in regulating various biological processes [30], including leaf senescence and plant cell death [31,32,33]. In research on autophagy in plants, it has been proven that the response to nutrition deficiency, senescence, intracellular degradation and recycling of amino acids, and other stresses from various environments can be regulated by autophagy [34,35]. Nitrogen is a crucial nutrient element required for plant growth, and autophagy, which is associated with plant aging, is activated as a response to nitrogen deprivation [36,37,38]. Nevertheless, it remains ambiguous whether DNA methylation plays a role in autophagy triggered by nitrogen deprivation. In previous research, *Arabidopsis* Col-0 and *atg5-1* mutant lines exhibited distinct phenotypic characteristics while growing under MS and nitrogen starvation conditions. Notably, the *atg5-1* genotype exhibited faster senescence and cell death under nitrogen starvation conditions, implying that autophagy potentially expedites senescence and cell death in comparison to Col-0-N. The dynamic changes in DNA methylation in Col-0-N and *atg5-1-*N corresponded to phenotype, wherein the number of hypo-DMRs was highest in the CG, CHG, and CHH contexts (Figure 1C), and among all the groups, 142 upregulated DEGs associated with hypo-DMRs were the most abundant in this comparison group (Figure 3A). The findings suggest that under conditions of nitrogen starvation, the *atg5-1* genotype may reduce the expression of methylated regions and potentially attract transcription factors to regulate gene expression and react to nitrogen starvation [39,40]. Based on the above results, the prediction of TFs that bind to hypo-DMRs was performed on Col-0-N vs. *atg5-1*-N, identifying nine TFs (*ERF105*, *ERF57*, *ATERF48*, *ERF111*, *AT3G46070*, *AT3G12910*, *ANAC004*, *ANAC087*, *ANAC046*) and five upregulated DEGs (*ATZAT12*, *ATMYB101*, *ATGH3.12*, *AT1G32763*, *LECRK-III.2*) binding with the above nine TFs from the GO terms response to biotic stimulus, cell death, catabolic process, and response to organonitrogen compound.

### 3.1. DNA Methylation Participates in Arabidopsis Senescence through the Regulation of ABA Biosynthesis

The process of PCD, known as senescence, has been found to be regulated by endogenous ethylene production under the control of specific transcription factors [41,42]. In our study, hypo-DMR-associated upregulated DEGs that might be regulated by TFs were identified, including *MYB101* (*AT2G32460*) and *ATGH3.12* (*AT5G13320*). Previous research indicates that *MYB101* functions as a constructive regulatory element in ABA-stimulated growth cessation, and an ineffective *MYB101* mutant reduces the sensitivity of plants to ABA [43,44]. It has been observed that ABA has the potential to induce senescence in leaves through a pathway that is independent of ethylene. Various scientific reports have indicated that the genes that encode rate-limiting enzymes involved in the biosynthesis of ABA include NCED2, NCED3, NCED5, NCED6, and NCED9 [45,46]. These five genes were differentially expressed (*p*-value < 0.1) in our analysis of gene expression differences between Col-0-N and *atg5-1*-N. Notably, NCED3 was identified as the most crucial rate-limiting enzyme (Figure 5B). It can be inferred that *atg5-1* induces high expression of *MYB101* under nitrogen starvation conditions, which consequentially exerts a positive regulatory effect on ABA synthesis and consequentially regulates *Arabidopsis* senescence. This regulatory relationship may be due to the downregulation of the methylation of the *MYB101* promoter region, which is achieved by recruiting TFs. Our investigation revealed that there are five TFs that have the capacity to bind to hypo-DMRs within the promoter region of the *MYB101* gene. Notably, these genes belong predominantly to the ERF family. Several research studies have demonstrated the participation of transcription factors from the ERF family in the deterioration and mortality mechanisms of plants. Based on research conducted by Chen, Y. et al., ERF.F5 plays a pivotal role in the senescence of tomato leaves. Specifically, the study indicated that ERF.F5 can directly regulate the promoter activity of *ACS6* and interact with *MYC2*, which ultimately leads to the regulation of tomato leaf senescence [47]. The transcription factor EPI1, also belonging to the ERF family, plays a negative regulatory role in dark-induced and JA-stimulated aging [48]. Furthermore, Juanxu Liu and colleagues discovered a robust correlation between numerous ERF genes and an increase in ethylene levels in petals and pistils during their examination of petunia blooms. This correlation is intimately linked with the process of petunia flower aging [49]. Therefore, we speculate that *ERF57*, *ERF105*, *ERF48*, and *ERF111* may be potential transcription factors that affect the aging of autophagic mutants under nitrogen starvation conditions. They are highly likely to bind to the hypo-DMR of the *MYB101* promoter, which affects the premature aging of the *atg5-1* mutant by upregulating the expression of *MYB101*.

### 3.2. SA Biosynthesis Was Probably Regulated by DNA Methylation

The ATGH3.12/PBS3 enzyme belongs to the GH3 acyl adenosylase enzyme family and plays a vital role in the accumulation of SA. Its main function is facilitating the conversion of isochorismate to unstable intermediates. Together with EPS1, it forms a two-step metabolic pathway in *Arabidopsis* to form SA, and this pathway is involved in the last two steps of SA formation [50]. SA plays an important role in regulating innate immunity and various developmental processes in plants. Research conducted by Niu, F. et al. showed that the transcription factor WRKY42 can regulate the aging process in *Arabidopsis* leaves by regulating the synthesis of SA [51,52]. According to reports, ICS1 is the key enzyme controlling SA synthesis, controlling 90% of this process [53,54]. Our study has demonstrated that there are significant changes in gene expression for two crucial enzymes, namely, *ATGH3.12*(*PBS3*) and *ICS1* (*p*-value < 0.05; FC > 1.5) (Figure 5), which means that SA biosynthesis is likely to be regulated, and DNA methylation may be involved in regulating the expression level of *ATGH3.12/PBS3*, thereby affecting SA biosynthesis. We also predicted a transcription factor (AT3G46070) that can probably bind to the hypo-DMR of the promoter of *ATGH3.12* (*PBS3*) and is differentially expressed; this transcription factor belongs to the C2H2-type zinc finger family of proteins.

Based on hormone level determination, our findings indicate that DNA methylation has a greater impact on SA biosynthesis, which probably contributes greatly to the premature senescence of *Arabidopsis* (Figure 7).

## 4. Materials and Methods

### 4.1. Plant Materials

*The Arabidopsis thaliana* Col-0 ecotype was used to elucidate methylation and transcription changes under nitrogen starvation and autophagy defects. The seeds were first cultivated in MS liquid culture medium for 7 days (16 h light/8 h darkness; constant temperature 22 °C; LED lighting; 120 µmol m^−2^ s^−1^) and then transferred into MS nitrogen starvation culture medium (MS-N) (MSP21-50LT) and cultivated for another 60 h (16 h light/8 h darkness; constant temperature 22 °C; LED lighting; 120 µmol m^−2^ s^−1^). The samples were dehydrated and snap-frozen in liquid nitrogen for subsequent WGBS, RNA-seq, and qRT–PCR analyses.

### 4.2. WGBS and Data Analysis

The DNA of the samples was extracted, and then an ultrasonic DNA fragmentation instrument was used to cleave the DNA sample (adding a certain proportion of Lambda DNA). A QSeq400 was used to verify cleavage (200–400 bp), and magnetic beads were used for purification. The purified product was subjected to bisulfite treatment and purification using the Method Gold Kit (ZYMO; Irvine, CA, USA). Finally, the product was amplified by PCR and sequenced using the Illumina platform at igenebook (www.ignenbook.com). The sequence of the sample was compared with that of lambda DNA to calculate the bisulfite conversion rate.

We used fastqc to evaluate raw data, and cutadapt [55] was used to remove low-quality bases from the 3′ ends, keeping the N base at no more than 5% of the total length and the length of reads at not less than 30 bp. In the mapping process, clean data were mapped to the *Arabidopsis thaliana* [56] reference genome by BSMAP [57]. The bisulfite conversion rate was calculated according to the result of mapping with lambda DNA. The coverage of cytosine bases across the genome and methylation level of three types of contexts (CG, CHG, CHH) were determined by CGmaptools. A PCA for each treatment group was performed according to the methylation level of each cytosine in the whole genome to evaluate the methylation level of each sample. The DMCs and DMRs were calculated using the CGmaptools dms and CGmaptools dmr tools (*p*-value < 0.001, ΔmC ≥ 0.2) [58]. For DMR-related genes, the differentially expressed genes (DEGs) identified from RNA-seq data (|log_2_FC| ≥ 2 and FDR < 0.01) and located within 2000 bp upstream or downstream of the DMR were defined as DMR-related genes. Gene Ontology (GO) and Kyoto Encyclopedia of Genes and Genomes (KEGG) pathway analyses were established by the Omicshare cloud platform with a Q-value cutoff of 0.05 (https://www.omicshare.com/tools/ accessed on 9 September 2023).

### 4.3. Transcriptome Sequencing (RNA-seq) and Data Analysis

Total RNA was extracted from each sample to prepare RNA libraries, and then RNA sequencing libraries were sequenced by IGENEBOOK Biotechnology Co., Ltd. (Wuhan, China) Genes with |log_2_FC| ≥ 2 and *p*-value < 0.05 were defined as differentially expressed genes (DEGs) [59]. The Q-values obtained by Fisher’s exact test were adjusted for multiple comparisons with the false discovery rate [60].

### 4.4. qRT–PCR and Determination of Hormone Content

Col-0 and *atg5-1* plants were grown in complete Murashige and Skoog (MS) medium for 1 week, which was then changed to −N or +N liquid medium. Sixty hours later, plants were harvested from different groups (three biological replicates). Total RNA was extracted by a FastPure Plant Total RNA Isolation Kit (Vazyme, Nanjing, China) and reverse transcribed to cDNA by HiScript III RT SuperMix (Vazyme) according to the manufacturer’s protocol. qPCR was performed using a CFX Connect PCR system (Bio-Rad, Hercules, CA, USA) according to the SYBR protocol (ChamQ Universal SYBR qPCR Master Mix, Vazyme), and the expression level was determined by the 2 − ΔΔCT method.

An amount of 50 mg of plant sample was weighed into a 2 mL plastic microtube, frozen in liquid nitrogen, and dissolved in 1 mL of methanol/water/formic acid (15:4:1, *v*/*v*/*v*). Ten microliters of internal standard mixed solution (100 ng/mL) were added to the extract as an internal standard (IS) for quantification. The mixture was vortexed for 10 min and then centrifuged for 5 min (12,000 r/min, 4 °C). The supernatant was transferred to a clean plastic microtube, evaporated to dryness, dissolved in 100 μL of 80% methanol (*v*/*v*), and passed through a 0.22 μM membrane filter for further LC–MS/MS analysis [61,62,63,64].

## 5. Conclusions

In this study, DNA methylation sequencing and RNA-seq sequencing of the whole genome of *Arabidopsis thaliana* showed that the expression of many genes was widely activated under nitrogen deficiency, especially when the autophagy function was defective, and the level of DNA methylation changed significantly. Under nitrogen deficiency, according to the analysis of DNA hypomethylation positions in autophagic mutants, many hypo-DMRs were found in the promoter regions of genes, which were largely related to the growth and death mechanisms of *Arabidopsis*. In our study, we found significant hypomethylation in the promoter regions of *MYB101* and *ATGH3.12* (*PBS3*). Based on the hypo-DMR sequence, we predicted transcription factors that may bind to the genes mentioned above (*ERF57*, *ERF48*, *ERF111*, *ERF105*, *and AT3G46070*) and verified differential upregulation of *MYB101*, *ATGH3.12* (*PBS3*), and related transcription factors through RNA-seq and qRT–PCR. *MYB101* and *ATGH3.12* (*PBS3*) have been confirmed to be associated with the biosynthesis of ABA and SA and are involved in regulating plant aging. Therefore, we applied LC–MS/MS technology to determine the hormone content of SA and ABA, and the results showed that under nitrogen-deficient growth conditions, the levels of SA and ABA in autophagic mutants were significantly upregulated compared to those in the wild type. By combining the results of WBGS, RNA-seq, and qRT–PCR, we speculated that under nitrogen deficiency, the premature senescence of the autophagy mutant of *Arabidopsis* was probably due to the change in DNA methylation level in the promoter region of *MYB101* and *ATGH3.12* (*PBS3*), leading to the recruitment of ERF family and C2H2-type zinc-finger family transcription factors (AT3G46070) to activate their own expression and then regulate ABA and SA to promote premature senescence.

In addition, in subsequent studies, we should focus on the molecular mechanism of autophagy affecting DNA methylation and regulating key genes in SA- and ABA-related pathways under nitrogen deficiency conditions and evaluate the impact of DNA methylation during plant growth.

## Figures and Tables

**Figure 1 ijms-24-14047-f001:**
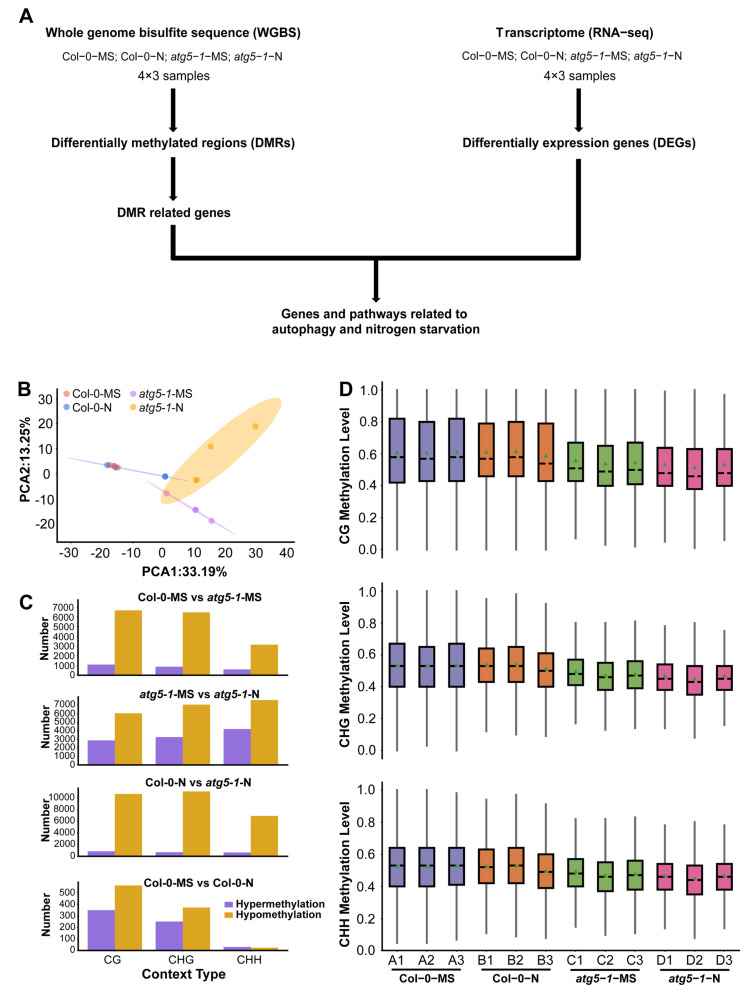
Autophagy function defects lead to changes in genome-wide methylation levels. (**A**) Data analysis workflow for whole-genome bisulfite sequencing (WGBS) and transcriptome analysis (RNA-seq). (**B**) PCA of the DNA methylation level in each sample. PCA of Col-0 and *atg5-1* genotypes, displaying obvious differences under MS and nitrogen starvation conditions. (**C**) The number of hypomethylated and hypermethylated DMRs in each comparison group. (**D**) The distribution of DNA methylation levels. The distribution plot of the methylation level of common sites showed that DNA methylation was affected by autophagy.

**Figure 2 ijms-24-14047-f002:**
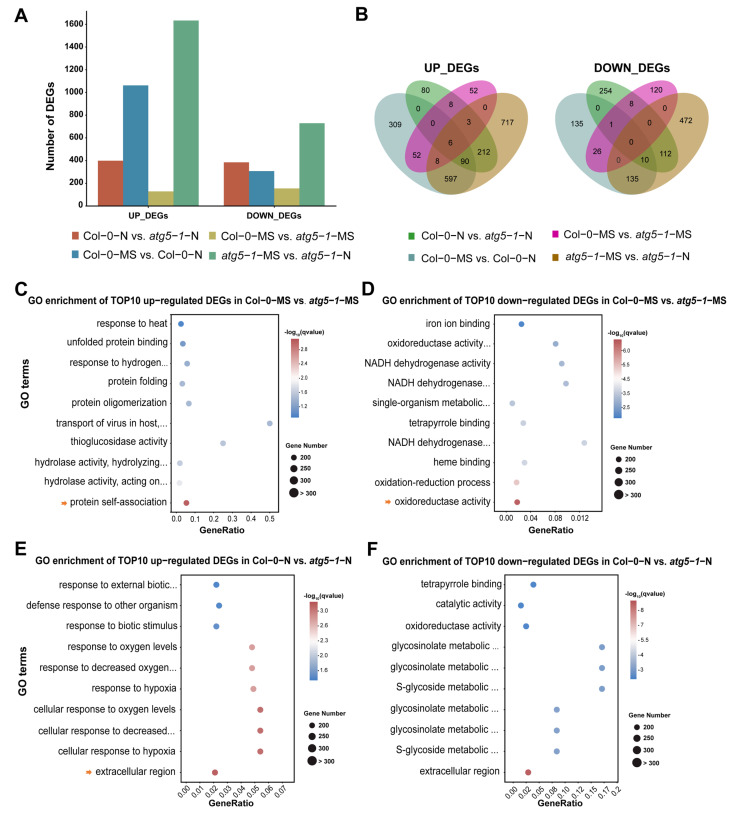
Nitrogen starvation activates differential gene expression across the whole genome of Col-0 and autophagic mutants. (**A**) DEG statistics. Gene expression was activated in the *atg5-1* and Columbia genotype groups under nitrogen starvation, showing the notable influence of nitrogen starvation. (**B**) Venn plot of DEGs in different comparison groups. (**C**–**F**) GO enrichment of the top 10 upregulated DEGs in Col-0-MS vs. *atg5-1*-MS, Col-0-MS vs. *atg5-1*-MS, Col-0-N vs. *atg5-1*-N, and Col-0-N vs. *atg5-1*-N. The arrow represents the GO terms with the most significant enrichment or that were meaningful for our research.

**Figure 3 ijms-24-14047-f003:**
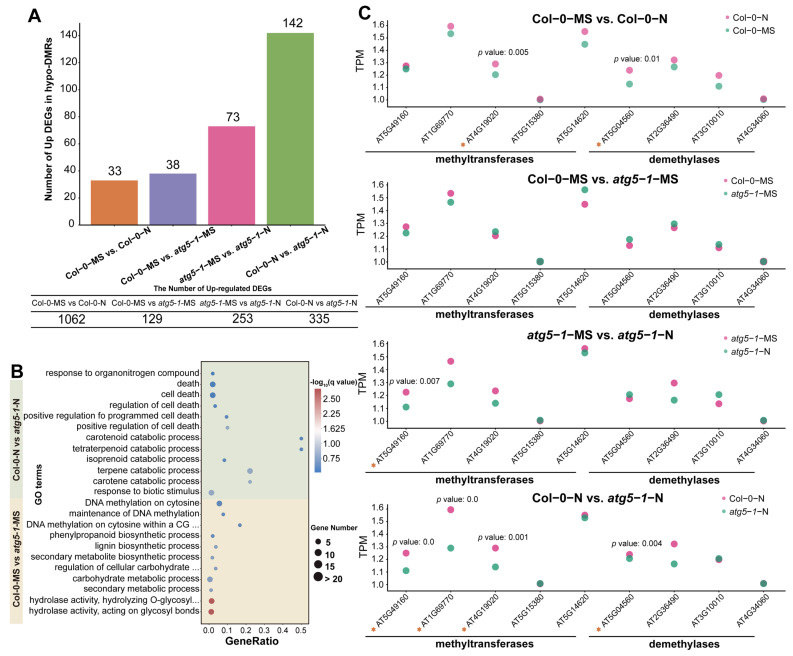
DNA methylation affects the response of autophagic mutants to nitrogen deficiency. (**A**) The number of hypo-DMRs associated with upregulated DEGs. For Col-0-N vs. *atg5-1*-N, half of the upregulated DEGs were associated with DMRs. (**B**) GO enrichment of hypo-DMRs associated with upregulated DEGs. (**C**) The expression of methyltransferases and demethylases calculated by RNA-seq indicated that DNA methylation was involved in the regulation of gene expression. The asterisks represent the transcription factors.

**Figure 4 ijms-24-14047-f004:**
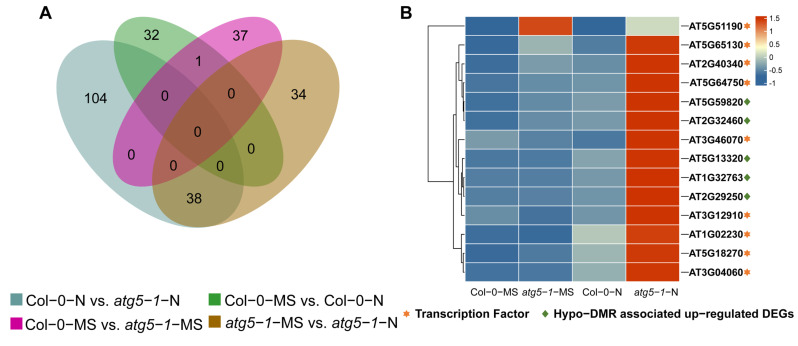
Senescence-related genes were strongly influenced by DNA methylation under nitrogen deficiency. (**A**) Venn plot for hypo-DMRs in different comparison groups. (**B**) Transcription factors (TFs) predicted to bind to the promoters of hypo-DMRs associated with upregulated DEGs. Most TFs belonged to the ERF family and NAC family, and most of them were highly expressed exclusively in *atg5-1*-N, which shows that these TFs and hypo-DMRs associated with upregulated DEGs probably play a key role in resisting nitrogen starvation in the *atg5-1* genotype. The asterisks and diamond represent the transcription factors and key genes related to plant death.

**Figure 5 ijms-24-14047-f005:**
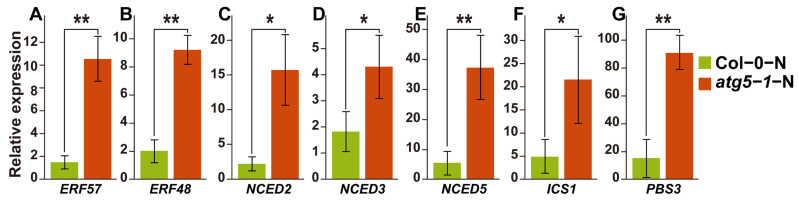
qRT–PCR results of key genes and predicted transcription factors that regulate the SA and ABA hormone pathways. ** Significant difference at *p* < 0.01 and * significant difference at *p* < 0.05. Vertical bars represent the standard deviation. (**A**–**E**) qRT–PCR of key genes in ABA biosynthesis pathway and predicted transcription factors that may be associated with ABA-induced growth arrest. (**F**,**G**) The expression levels of key genes regulated by DNA methylation in the SA biosynthesis pathway.

**Figure 6 ijms-24-14047-f006:**
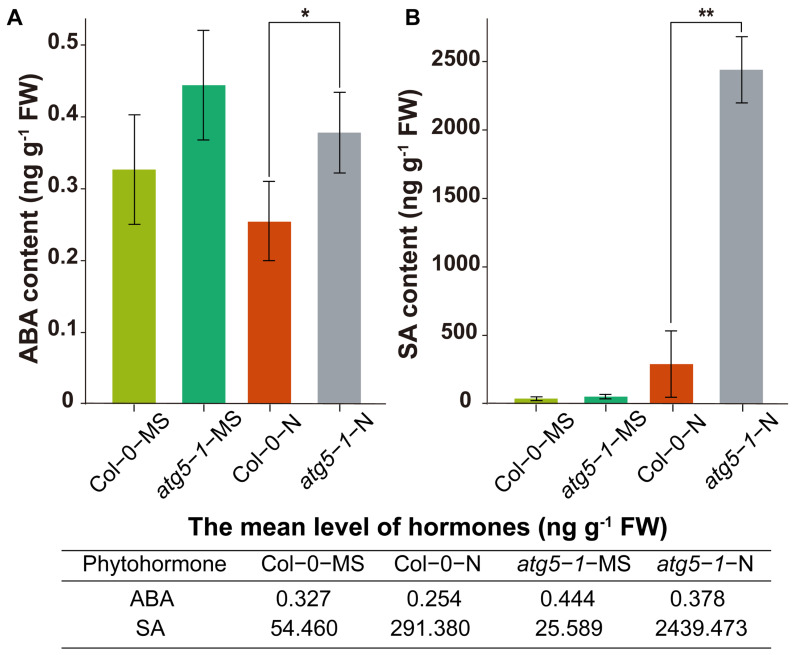
Quantification of the phytohormones ABA and SA. ** Significant difference at *p* < 0.01 and * significant difference at *p* < 0.05. Vertical bars represent the standard deviation. The levels of the hormones ABA (**A**) and SA (**B**) in the autophagic mutant were significantly higher than those in the wild type under nitrogen starvation.

**Figure 7 ijms-24-14047-f007:**
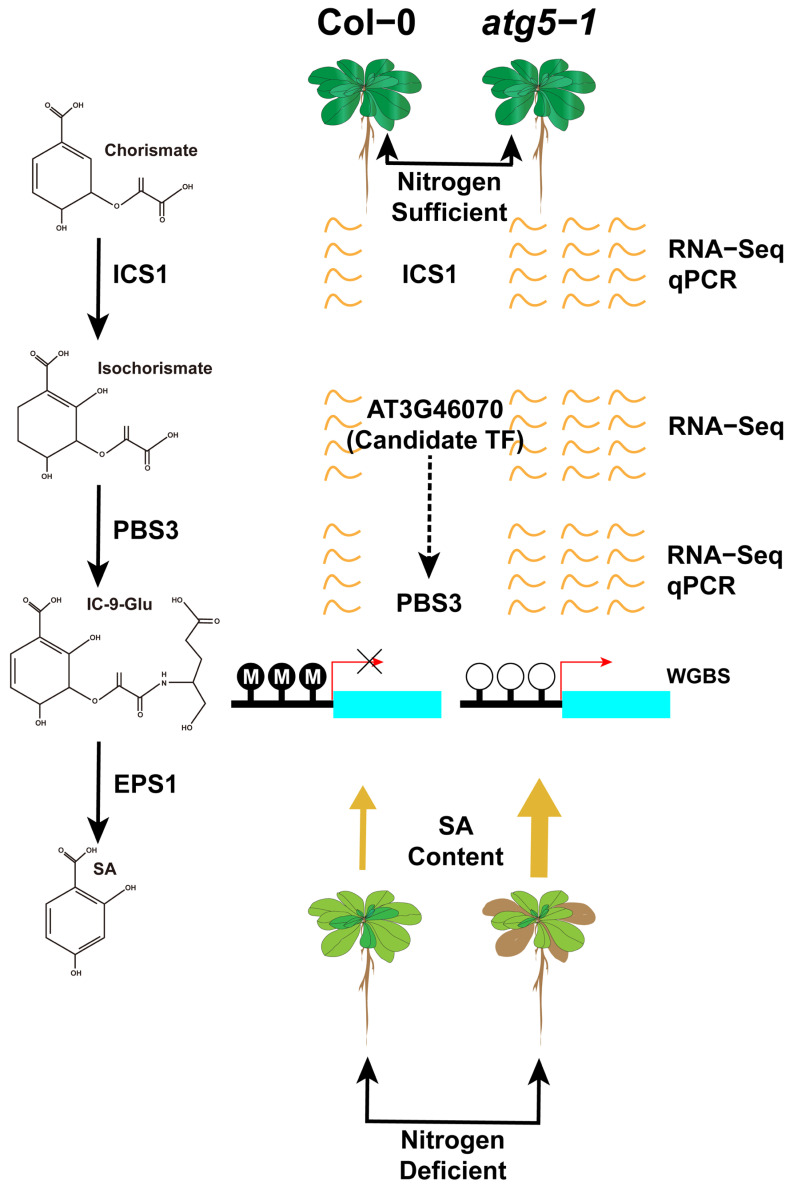
DNA methylation may regulate premature aging of *atg5-1* mutants under nitrogen starvation conditions by mediating the expression level of *PBS3*. By predicting transcription factors in the regions where hypomethylation occurs in the *PBS3* promoter, it is speculated that *AT3G46070* expressing transcription factors may be recruited to the hypomethylation regions, leading to an upregulation of PBS3 expression levels.

## Data Availability

The data presented in this study are available upon request from the corresponding author. The raw sequence data reported in this paper have been deposited in the Genome Sequence Archive in National Genomics Data Center, China National Center for Bioinformation/Beijing Institute of Genomics, Chinese Academy of Sciences (GSA: CRA012212 for WGBS; GSA: CRA012214 for RNA-seq) that are publicly accessible at https://ngdc.cncb.ac.cn/gsa/s/InLHU0Ef and https://ngdc.cncb.ac.cn/gsa/s/W6sx307x, accessed on 9 September 2023.

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
