# Peer review of "The Change in Whole-Genome Methylation and Transcriptome Profile under Autophagy Defect and Nitrogen Starvation"

_ijms, 2023, doi:10.3390/ijms241814047_

Round 1
Reviewer 1 Report
Title: The change of whole genome methylation and transcriptome profile under
autophagy defect and nitrogen starvation
Dear Authors
The subject is very interesting and fall within the scope of the journal. The experimental dataset undoubtedly are useful and constitutes scientific values.
In this study, whole-genome bisulfite sequencing (WGBS) was performed on Arabidopsis autophagic mutants (atg5-1) developing under nitrogen starvation to address relevant concerns , in conjunction with transcriptome sequencing (RNA-seq).
The manuscript is written correctly and does not raise any major objections.
Genaral Remarks
1. Line 30 WGBS – When using the abbreviation for the first time, the full name must also be provided.
2. Figure 1B, 1C, and 3B, and others – The readability of the description should be improved - the font should be increased.
3. Figure 6 - Units and their notation should be adapted to editorial requirements.
4. The discussion of the results is well written.
5. Conclusion - Directions for further research should be given.
6. The References section should be adapted to the publishing requirements. Please complete the doi numbers.
Best regards
Reviewer 2 Report
Research is relevant and interesting.
1. No research hypothesis is presented in the introduction.
2. In the Figures it is not explain what the asterisks and vertical bars represent.
3. The conclusions could be more concrete.
Reviewer 3 Report
Main comments:
The authors of the manuscript addressed the interesting topic of nitrogen starvation in plants in relation to their genome. The content is very important and useful in understanding the mechanisms of autophagy. The results obtained are promising and may be very useful in further research in this field. The possibilities of using such detailed research are very large, they can be used for further plant breeding and improvement of plant tolerance to environmental stresses. The manuscript is well written with detailed descriptions and figures.
Detailed comments and suggestions:
Results
Line 403: Figure 7. At the bottom of the picture, 'nitrogen deficient' is repeated and above you can see a diagram and 2 plants showing different symptoms of nitrogen deficiency. They should be signed, why there are these differences, specify these plants, and under them it is enough to specify one that concerns the nitrogen deficit.
Materials and Methods
Line 500: What were the day and night temperatures during plant growth? How intense was the light and what was the type of lighting? (LED, fluorescent).
